

# Sleep health of Australian community tennis players during the COVID-19 lockdown

Philipp Beranek[1,2,*], Travis Cruickshank[1,2,3,*], Olivier Girard[4], Kazunori Nosaka[2,5], Danielle Bartlett[1,2] and Mitchell Turner[1,2]

[1] Centre for Precision Health, Edith Cowan University, Joondalup, WA, Australia
[2] School of Medical and Health Sciences, Edith Cowan University, Joondalup, WA, Australia
[3] Perron Institute for Neurological and Translational Science, Edith Cowan University, Joondalup, WA, Australia
[4] School of Human Sciences (Exercise and Sport Science), University of Western Australia, Perth, WA, Australia
[5] Centre for Exercise and Sports Science Research, Edith Cowan University, Joondalup, WA, Australia
* These authors contributed equally to this work.

Corresponding author
Philipp Beranek,
p.beranek@ecu.edu.au

## ABSTRACT

**Background:** Poorer sleep health outcomes have been documented in the general population during the COVID-19 outbreak. However, the impact of the COVID-19 outbreak on sleep health outcomes in specific population groups, including the sporting community, has not been extensively investigated. This study evaluated sleep health outcomes and their relationship with lifestyle behaviours during the initial COVID-19 lockdown period in Australian community tennis players.
**Methods:** This cross-sectional study evaluated sleep health outcomes and lifestyle behaviours using an online survey. The survey was disseminated online between the 24th of April and the 6th of June 2020 and comprised the Sleep Health Index, Sleep Satisfaction Tool and questions regarding weekly hours of tennis play, general physical activity, training location and alcohol consumption. Two-hundred and eighty-five individuals completed the survey.
**Results:** Compared to normative data, respondents displayed positive sleep health values during the initial COVID-19 lockdown period, with median values (IQR) of 85.3 (73.4, 91.7) and 64.8 (54.4, 73.4) for the Sleep Health Index and Sleep Satisfaction Tool, respectively. Sleep health outcomes were not significantly correlated ($p > 0.05$) with tennis play (Tb = 0.054–0.077), physical activity (Tb = −0.008 to 0.036), training location (Tb = −0.012 to −0.005) or alcohol consumption (Tb = −0.079 to −0.018).
**Conclusion:** Positive sleep health values were observed in Australian community-level tennis players during the initial COVID-19 pandemic. Sleep health values were not associated with lifestyle behaviours. Other unexplored factors may have influenced sleep health outcomes, including personal finances and socialisation, however these factors need to be investigated in future studies.

## INTRODUCTION

Accumulating evidence indicates that sleep has been, and continues to be, negatively impacted by the COVID-19 pandemic and associated government-imposed restrictions (*Li et al., 2020*; *Robillard et al., 2021*; *Stanton et al., 2020*). An increased prevalence of insomnia symptoms, poorer self-reported sleep quality, later bedtimes, increased total time spent in bed and decreased sleep quality were reported during this period in adults residing in China (*Zhang & Xiao, 2020*) and Italy (*Casagrande et al., 2020*). While informative, these studies have mainly used questionnaires that are designed to evaluate symptoms of sleep conditions rather than overall sleep health.

Sleep health is not merely an absence of sleep disorders, but rather "a multidimensional pattern of sleep-wakefulness, adapted to individual, social, and environmental demands, that promotes physical and mental well-being" (*Buysse, 2014*). To holistically evaluate sleep health, the National Sleep Foundation (NSF) developed and validated the Sleep Health Index (SHI) and Sleep Satisfaction Tool (SST) and provide normative data for the general population (*Knutson et al., 2017*; *Ohayon et al., 2019*). Only a handful of studies have investigated sleep health during the pandemic using validated sleep health questionnaires (*Coelho et al., 2021*; *Targa et al., 2021*; *Yuksel et al., 2021*). *Yuksel et al. (2021)* reported poor sleep health values in 44.4% of surveyed healthy adults ($n$ = 6,882, 18–94 years of age, 78.8% females) from 59 countries, indicated by RU-SATED scores below the median of 7.0. In this study, stricter quarantine was associated with poorer sleep outcomes, which was attributed to decreased time spent outdoors, less physical activity and reduced daylight exposure (*Yuksel et al., 2021*). However, there was considerable heterogeneity in this sampled population, particularly with respect to COVID-19 infection and death rates and the lifestyle behaviours. These factors may have influenced respondents' sleep health, therefore limiting the strength of conclusions that could be derived. Examining sleep health during the pandemic in specific populations and its association with lifestyle behaviours is therefore of importance.

Existing studies have noted changes in lifestyle behaviours during the COVID-19 pandemic, with numerous studies reporting a negative change in physical activity in the general community (*Castañeda-Babarro et al., 2020*; *Lesser & Nienhuis, 2020*; *Stanton et al., 2020*) and athletes (*Facer-Childs et al., 2021*; *Washif et al., 2021*). In particular, *Stanton et al. (2020)* reported a negative change in physical activity in 48.9% of sampled Australian adults ($n$ = 1,491). In elite and semi-elite athletes, *Facer-Childs et al. (2021)* observed a disruption in physical activity in 78.9% of their sample ($n$ = 565). Furthermore, several studies have reported an increase in alcohol consumption in the general community (*Barbosa, Cowell & Dowd, 2021*; *Neill et al., 2020*) and athletes (*Facer-Childs et al., 2021*; *Imboden et al., 2021*) during the pandemic. Specifically, *Neill et al. (2020)* observed in 30.8% of sampled Australian adults ($n$ = 4,462) reporting "drinking a lot more than normal". Similar findings were observed in Australian elite and semi-elite athletes by *Facer-Childs et al. (2021)*, with 25% of their sample ($n$ = 565) reporting increased alcohol consumption. These preliminary findings are of interest given the previously reported link between physical activity and alcohol consumption, with sleep health.

In particular, regular physical activity (*Kredlow et al., 2015*; *Rubio-Arias et al., 2017*) and lower alcohol consumption (*Chueh, Guilleminault & Lin, 2019*) are associated with superior sleep efficiency and quality in adults. Despite the adverse impact of the pandemic on various aspects of sleep health, including sleep quality and timing (*Li et al., 2020*; *Robillard et al., 2021*; *Stanton et al., 2020*), only a handful of studies have explored the role of lifestyle behaviours during this period, including changes to physical activity (*Robillard et al., 2021*; *Yuksel et al., 2021*) and alcohol consumption (*Ingram, Maciejewski & Hand, 2020*; *Maugeri et al., 2020*) on sleep health (*Chodkiewicz et al., 2020*). *Stanton et al. (2020)* observed that decreased physical activity was related to poorer sleep outcomes in the Australian community. It is therefore conceivable that individuals engaging in community level sports may have better sleep health outcomes during the pandemic. To our knowledge, sleep health has not been investigated in individuals engaging in community-level sports.

Tennis is played by 1.12% (~87 million people) of the population world-wide and 6.2% (~1.12 million people) within Oceania, making it one of the most popular sports (*International Tennis Federation, 2019*). Tennis is also one of the few sports that is played equally by both females (47%) and males (*International Tennis Federation, 2019*). Tennis is played at various ages, thus allowing young, middle aged and older adults to remain physically active. At the beginning of the initial COVID-19 lockdown in Australia, sporting facilities were forced to close. This led to restricted training hours for athletes, with *Facer-Childs et al. (2021)* reporting a significant reduction in training duration and weekly training frequency. Tennis facilities were among the first sporting venues to reopen as players were able to maintain physical distance while playing. Therefore, tennis represents a large and diverse sample to investigate the sleep health and lifestyle behaviours of community athletes during the pandemic.

The aim of this study was twofold, (1) to evaluate sleep health during the initial COVID-19 lockdown period (24th of April–6th of June 2020) in Australia and (2) to assess its association with tennis play, physical activity, training location (indoors, outdoors), and alcohol consumption in Australian adult community-level tennis players. We hypothesised that sleep health would be lower than established normative values and associated with less physical activity and tennis play, as well as greater alcohol consumption.

## MATERIALS AND METHODS

### Study design

A cross-sectional study design was used to explore sleep health and its contributing factors during the COVID-19 pandemic in community-level tennis players. Sleep health, number of hours of tennis play and physical activity per week, training location, and alcohol consumption during this period were evaluated using an online survey that was administered *via* Qualtrics. The survey link was disseminated within Australia *via* social media (*e.g.*, Twitter, Facebook, ResearchGate and LinkedIn) and was active between the 24th of April and the 6th of June 2020. During this timeframe, facilities (*e.g.* sporting facilities and cinemas) were temporarily closed or restricted due to government-imposed

rules and physical distancing measures were mandatory. This study was approved by the Edith Cowan University Human Research Ethics Committee (2020-01367).

## Participants

Participants were community-level tennis players from Australia. Participants could access the online survey *via* a link and were provided with the study information sheet to read and download. All participants gave written informed consent before they completed the survey. Inclusion criteria included; (1) regular (at least once per week) tennis play, (2) have residency in Australia and (3) be 21 years of age or above. The number of hours performing physical activity other than tennis was recorded and included in the analysis to control for any effect on sleep health.

## Measures

### Demographics

Demographic information included sex, age, tennis experience, match format, relationship status and if survey respondents had contracted COVID-19. The age ranges of participants were collected as either 21–30 years, 31–40 years, 41–50 years, 51–60 years, 61–70 years, and 71 years and above. Furthermore, tennis experience was defined as years playing tennis and match format as singles and/or doubles.

### Sleep health

Sleep health was assessed using two measures developed by the National Sleep Foundation; the Sleep Health Index (SHI) and the Sleep Satisfaction Tool (SST).

The SHI is a validated and reliable measure designed to assess multiple dimensions of sleep health within the general population (*Knutson et al., 2017*). Construct validity was demonstrated by significant ($p < 0.001$) correlations between SHI and respondent's ratings of overall health ($r = 0.38$), stress ($r = −0.37$), and life satisfaction ($r = 0.36$) (*Knutson et al., 2017*). Prior work by the National Sleep Foundation has shown that the SHI has a Cronbach $\alpha$ 0.76 (*Knutson et al., 2017*). It consists of 12 items that assess the domains of sleep quality, sleep duration, and disordered sleep (see Fig. S1). Sleep quality is evaluated by the participant's rating of six items, including their sleep quality, number of days out of the last seven that they felt well-rested, trouble falling asleep, trouble staying asleep, the number of days out of the last seven that poor sleep negatively impacted performance of daily activities, and the number of days out of the last seven that they unintentionally dozed. Sleep duration is evaluated by the participant's rating of three items, including the usual weekday sleep times of the past 7 days, sleep deficit as the difference between the amount of sleep on weekdays and the amount of sleep the respondents need to feel at their best, and sleep variability based on weekday time in bed and weekend time in bed of the past 7 days. Disordered sleep is evaluated by the participant's rating of three items, including the number of nights out of the last seven that the participants took sleep medication, if a sleep disorder has been diagnosed, and if they had discussed any sleep problems with a sleep doctor. Each dimension results in a sub-index score based on its items' rating. The mean of all three sub-index scores represents the overall SHI score

that ranges from 0–100 with a higher and lower number indicating better and worse sleep health, respectively.

The SST is a subjective measure of sleep satisfaction that gives an indication of sleep health and wellness independent of sleep quality or quantity (*Ohayon et al., 2019*). Convergent validity was demonstrated by significant ($p < 0.001$) correlations between SST and respondent's ratings of overall health (r = 0.40), stress (r = −0.40), and life satisfaction (r = 0.37) (*Ohayon et al., 2019*). The SST has previously been reported to have strong reliability as measured by Cronbach α (0.87, $R^2 = 0.5$) (*Ohayon et al., 2019*). The SST score is based on the participants' rating of nine items including sleep satisfaction, feeling refreshed, weekday sleep satisfaction, weekend sleep satisfaction, feeling energized, trouble falling asleep, achieving relaxed mental state, falling back asleep, and waking up during the night (see Fig. S2). The SST score ranges from 0–100 with a higher and lower number indicating higher and lower sleep satisfaction, respectively.

Currently there are no clinical cut-off scores for SHI and SST available. Therefore, the SHI and SST scores, reported in the present study, are compared to normative values measured by *Knutson et al. (2017)* and *Ohayon et al. (2019)* in healthy American adults prior to the COVID-19 pandemic. Higher scores compared to the normative values are interpreted as positive sleep health in the present study, while lower scores are interpreted as negative sleep health.

### Tennis play, physical activity, and training location

To evaluate tennis play and physical activity, the survey included questions regarding the number of weekly hours playing tennis and performing physical activity (other than tennis). Furthermore, the respondents were asked about the location they performed their physical training during the pandemic (see Fig. S3).

### Alcohol consumption

Information regarding alcohol consumption was assessed with this survey. Based on their frequency of consuming alcohol since the COVID-19 situation, the respondents selected one of the following choices: "I don't drink alcohol at all", "less than once a month", "once a month", "once every two weeks", "once every week", or "few times a week" (see Fig. S3).

### Statistical analysis

A Shapiro–Wilk test was performed to check the data for normality. Data were not normally distributed. Therefore, data are presented as median and interquartile ranges (IQR). In the analysis of hours of tennis play and physical activity, participants who did not perform any physical activity (*e.g.*, tennis play) during the pandemic, were excluded. Participants' training locations were classified as outdoors when at least one outdoor location was reported, otherwise classified as indoors. Sex and age group differences in SHI, SST, tennis play, physical activity, training location, and alcohol consumption were assessed using a Mann–Whitney U test and a Kruskal–Wallis test, respectively. One participant that did not identify as male, female, or transgender, was removed for this analysis. A Kendall's Tau B Correlation analysis was performed to evaluate potential

associations between sleep health measures and age, tennis play, physical activity, training location, and alcohol consumption. The *Jamovi Project (2021)* was used for statistical analysis was used for statistical analysis. Significance was set at <0.05.

## RESULTS

### Participants

As seen in Table 1, 285 respondents met the inclusion criteria. The majority of respondents ($n = 81$) were aged 51–60 years, while the lowest numbers ($n = 20$) were reported in the two youngest age groups: 21–30 years and 31 to 40 years, respectively. The respondents included 156 males (54.7%) and 128 females (44.9%), with one respondent (0.4%) not identifying as male, female, or transgender. The majority (78.2%) of participants had been playing tennis for more than 20 years, with only 2.1% of respondents playing tennis for 1–3 years. The majority (81.1%) were in a committed relationship. Only 0.4% of the respondents had contracted COVID-19, whilst 4.9% were unsure.

### Sleep health

As shown in Fig. 1, the median SHI score was 85.3 (73.4, 91.7). Median sub-index scores were as follows: 97.5 (88.6, 100) for sleep duration, 70.2 (56.0, 82.0) for sleep quality and 100 (66.7, 100) for disordered sleep. The median SST score was 64.8 (54.4, 73.4). Females had significantly higher disordered sleep sub-index scores compared to males ($p = 0.01$, d = 0.27), but no other differences were observed between males and females for sleep health measures (SHI ($p = 0.10$, d = 0.19), sleep duration ($p = 0.08$, d = −0.17), sleep quality ($p = 0.24$, d = 0.17), and SST ($p = 0.11$, d = 0.18). No significant differences were found for median scores of the SHI ($p = 0.26$, $X^2 = 6.50$), sleep duration ($p = 0.55$, $X^2 = 4.01$), sleep quality ($p = 0.87$, $X^2 = 1.82$), disordered sleep ($p = 0.14$, $X^2 = 8.26$) or SST ($p = 0.87$, $X^2 = 1.88$) between age groups.

### Tennis play, physical activity, and training location

The number of hours respondents performed tennis play and physical activity, the percentage of respondents performing physical activity indoors and outdoors and the percentage of respondents not performing any physical activity are displayed in Table 1. No significant difference in the number of hours per week that participants partook in tennis play was found between sex ($p = 0.26$, d = 0.10) or age groups ($p = 0.28$, $X^2 = 6.33$). During the pandemic, 41% of respondents performed at least some physical activity outdoors. There was no significant difference in weekly hours of physical activity between sex ($p = 0.54$, d = −0.05) or age groups ($p = 0.18$, $X^2 = 7.68$). No significant difference in training location for sex ($p = 0.78$, d = −0.03) or age groups ($p = 0.69$, $X^2 = 3.10$) was found.

### Alcohol consumption

As displayed in Table 1, more than half (56.5%) of the cohort reported drinking alcohol a few times a week, whilst 9.5% did not drink any alcohol during the COVID-19 pandemic. No significant difference was found in frequency of alcohol consumption between sex ($p = 0.83$, d = −0.01) or age groups ($p = 0.22$, $X^2 = 7.00$).

**Table 1  Demographics and lifestyle factors of participants during the pandemic.**

| | Totals (n = 285) | | Males (n = 156, (54.7%)) | | Females (n = 128, (44.9%)) | |
|---|---|---|---|---|---|---|
| **Age range (years)** | | | | | | |
| 21–30 | 7.0% | (20) | 9.0% | (14) | 4.7% | (6) |
| 31–40 | 7.0% | (20) | 9.6% | (15) | 3.9% | (5) |
| 41–50 | 17.9% | (51) | 19.9% | (31) | 14.8% | (19) |
| 51–60 | 28.4% | (81) | 24.4% | (38) | 33.6% | (43) |
| 61–70 | 27.7% | (79) | 22.4% | (35) | 34.4% | (44) |
| 71+ | 11.9% | (34) | 14.7% | (23) | 8.6% | (11) |
| **Relationship status** | | | | | | |
| Single | 17.2% | (49) | 16.0% | (25) | 18.8% | (24) |
| In relationship | 81.1% | (231) | 84.0% | (131) | 77.3% | (99) |
| Other | 1.8% | (5) | – | | 3.9% | (5) |
| **Tennis experience (years)** | | | | | | |
| 1–3 | 2.1% | (6) | 1.3% | (2) | 3.1% | (4) |
| 3–5 | 3.2% | (9) | 3.8% | (6) | 2.3% | (3) |
| 5–10 | 4.2% | (12) | 5.1% | (8) | 3.1% | (4) |
| 10–15 | 5.6% | (16) | 5.8% | (9) | 5.5% | (7) |
| 15–20 | 6.7% | (19) | 7.1% | (11) | 6.3% | (8) |
| >20 | 78.2% | (223) | 76.9% | (120) | 79.7% | (102) |
| **Match format** | | | | | | |
| Singles | 5.6% | (16) | 9.0% | (14) | 1.6% | (2) |
| Doubles | 68.8% | (196) | 56.4% | (88) | 83.6% | (107) |
| Singles and doubles | 25.6% | (73) | 34.6% | (54) | 14.8% | (19) |
| **Have you contracted COVID-19** | | | | | | |
| Yes | 0.4% | (1) | 0.3% | (1) | – | |
| No | 94.7% | (270) | 60.7% | (149) | 94.5% | (121) |
| Unsure | 4.9% | (14) | 3.1% | (6) | 5.5% | (7) |
| **Tennis play and physical activity (hours)** *Presented as median (interquartile ranges)* | | | | | | |
| Tennis play | 1.0 (0.0, 2.0) | | 1.0 (0.0, 2.0) | | 0.0 (0.0, 2.0) | |
| Physical activity | 4.0 (2.0, 8.0) | | 4.0 (2.0, 8.0) | | 4.0 (2.0, 8.0) | |
| **Training location** | | | | | | |
| Indoors | 36.1% | (103) | 37.2% | (58) | 35.2% | (45) |
| Outdoors | 40.7% | (116) | 39.7% | (62) | 41.4% | (53) |
| No physical activity | 23.2% | (66) | 23.1% | (36) | 23.4% | (30) |
| **Alcohol consumption** | | | | | | |
| I don't drink alcohol at all | 9.5% | (27) | 11.5% | (18) | 7.0% | (9) |
| Less than once a month | 10.2% | (29) | 8.3% | (13) | 12.5% | (16) |
| Once a month | 4.6% | (13) | 5.1% | (8) | 3.9% | (5) |
| Once every 2 weeks | 6.0% | (17) | 4.5% | (7) | 7.8% | (10) |
| Once a week | 13.3% | (38) | 12.8% | (20) | 14.1% | (18) |
| Few times a week | 56.5% | (161) | 57.7% | (90) | 54.7% | (70) |

**Note:**
Data presented as percentage (total number) unless stated otherwise.

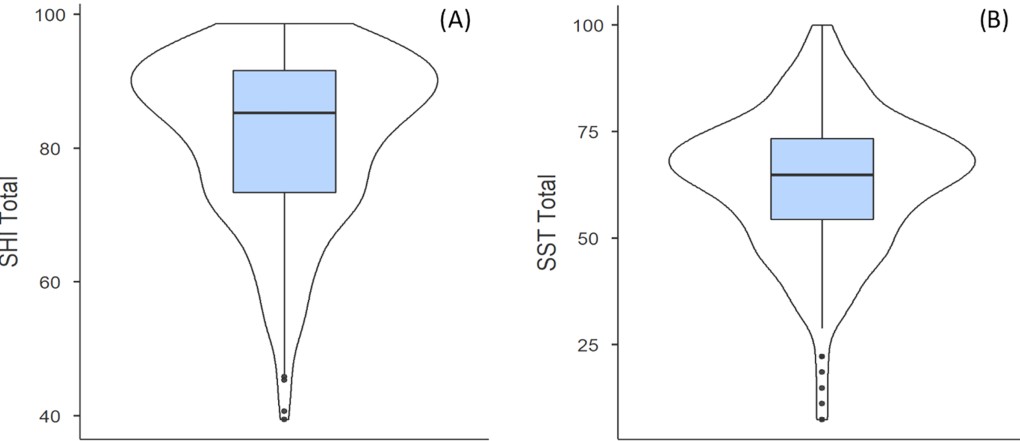

**Figure 1 Sleep Health Index (A) and Sleep Satisfaction Tool scores (B).** The median for each group is indicated by the thick black line, with the interquartile ranges shown by the light blue box. The remaining distribution, excluding outliers, are shown by the thin black line, with outliers indicated by the black dots. The curved outer lines indicate the number of respondents with that particular score.

**Table 2 Associations between sleep health measures and age, and lifestyle behaviours during the COVID-19 pandemic.**

|  |  | Sleep Health Index | | | | Sleep Satisfaction Tool |
|---|---|---|---|---|---|---|
|  |  | **Total** | **Duration** | **Quality** | **Disorders** | **Total** |
| Age | $T_b$ | −0.08 | −0.00 | −0.02 | −0.10 | 0.02 |
|  | p | (0.06) | (0.99) | (0.63) | (**0.04**) | (0.72) |
| Tennis play | $T_b$ | 0.05 | −0.01 | 0.05 | 0.03 | 0.08 |
|  | p | (0.25) | (0.91) | (0.29) | (0.59) | (0.10) |
| Physical activity | $T_b$ | −0.04 | −0.05 | −0.03 | −0.03 | −0.01 |
|  | p | (0.42) | (0.27) | (0.49) | (0.61) | (0.85) |
| Training location | $T_b$ | −0.01 | −0.09 | −0.02 | 0.04 | −0.01 |
|  | p | (0.91) | (0.08) | (0.62) | (0.43) | (0.79) |
| Alcohol consumption | $T_b$ | −0.02 | 0.06 | −0.04 | −0.02 | −0.08 |
|  | p | (0.69) | (0.25) | (0.43) | (0.75) | (0.08) |

**Note:**
Significant ($p < 0.05$) associations in bold.

## Associations

As shown in Table 2, the number of hours playing tennis and performing physical activity did not significantly correlate with SHI or SST. Additionally, neither the training location nor the frequency of alcohol consumption were significantly correlated with SHI or SST. There was a significant ($p = 0.04$) negative ($T_b = -0.10$) association between age and the sub-index score "Disorders" of the SHI. No other significant association was found between age and any SHI sub-index scores or SST.

## DISCUSSION

The present study examined sleep health and its relationship with lifestyle behaviours in adult community-level tennis players during the COVID-19 pandemic in Australia. Compared to normative values, Australian community level tennis players exhibited positive sleep health values during the pandemic, indicated by higher scores on the SHI (normative: 76 *vs.* tennis players: 85.3) and SST (normative: 56 *vs.* tennis players: 64.8) (*Knutson et al., 2017*; *Ohayon et al., 2019*). There was a significant negative association between age and the SHI-sub-index score for sleep disorders, which is not surprising given a higher prevalence of sleep disorders observed with an increase in age (*Hossain & Shapiro, 2002*; *Wolkove et al., 2007*). Sleep health was not associated with engagement in tennis play, physical activity, training location or alcohol consumption.

A recent investigation by *Facer-Childs et al. (2021)* noted a negative impact of COVID-19 on sleep in elite and sub-elite athletes in Australia. In particular, authors noted a 25% increase in sleep latency as well as greater daytime sleepiness ($p = 0.01$), indicating poorer sleep outcomes. Our results contrast these earlier findings, specifically we observed positive sleep health values in the sampled group of Australian adult community-level tennis players during the pandemic. Several reasons may account for this discrepancy in finding. First, the present study only included community-level athletes, whereas *Facer-Childs et al. (2021)* recruited elite athletes (41% of participants), who are reliant on competitions to maintain or enhance their ranking and for generating income. Considering that elite sporting competitions were suspended during the study period (24th of April-6th of June 2020) in Australia, it is likely that elite athletes experienced greater emotional burden as their income was likely to have been negatively impacted, which may have adversely impacted their sleep health. Second, the majority (79%) of the respondents in *Facer-Childs et al. (2021)* were team sport athletes. Whilst no total lockdown was imposed by the Australian government, some restrictions (*e.g.*, physical distancing and limited number of people) were put in all states in place. As opposed to team sports, individual sports such as tennis could adhere to those government-imposed restrictions, therefore enabling them to engage in greater physical activity through tennis play. Thus, the sleep health of community-level tennis players may not have been as negatively impacted by the pandemic as other sub-elite team sports or elite athletes.

During the pandemic, Australian community-level tennis players performed a median of 1.0 h tennis play and 4.0 h of physical activity, which exceeds the World Health Organisation's recommendation of 150–300 min (2.5–5.0 h) of moderate intensity or 75–150 min (1.25–2.5 h) of vigorous intensity physical activity per week (*Bull et al., 2020*). Based on existing literature (*Kredlow et al., 2015*; *Rubio-Arias et al., 2017*), we hypothesised that a greater level of physical activity would be associated with better sleep health outcomes. Contrary to our expectations, we did not observe any association between physical activity levels and sleep health outcomes in the present study. This observation contradicts existing meta-analyses where regular physical activity was associated with better sleep outcomes, particularly sleep quality (*Banno et al., 2018*; *Kredlow et al., 2015*; *Semplonius & Willoughby, 2018*). The absence of a significant association between level
of physical activity and sleep health could be explained by the potential presence of a ceiling effect; overall, the tennis players might have been performing a sufficient level of physical activity that did not influence their sleep health. This supposition is supported given the majority of participants met the WHO recommendations for physical activity.

Exposure to natural light, which is a known zeitgeber, may influence the sleep health of Australian community tennis players. Extant studies have shown that sunlight exposure is linked to the quality and timing of sleep, two fundamental aspects of sleep health (*Düzgün, 2017*; *Wright et al., 2013*). *Facer-Childs et al. (2021)* reported a reduction of outdoor light exposure in 70% of the sampled elite and sub-elite Australian athletes, which may explain the noted disruption in sleep timing observed. Contrary to our expectations, training location was not correlated with overall or specific domains of sleep health. It may be that respondents were able to attain enough outdoor light exposure when they were not performing physical activity while also maintaining good sleep hygiene (*e.g.* limit screen time before bed) throughout the lockdown period.

Contrary to our expectations, alcohol consumption was not associated with sleep health outcomes. This is surprising given the consistent negative association reported between sleep and alcohol consumption (*Hu et al., 2020*). The lack of association between alcohol consumption and sleep health may, at least in part, be due to the measurement approach used in the present study, particularly the response items utilised. The present study asked respondents to indicate their frequency of alcohol consumption during the lockdown period. While informative, this survey item and the associated responses do not enable alcohol volume to be calculated. It is possible that alcohol volume, measured as millilitres or number of standard drinks consumed per week and occasion, may have been associated with sleep health, which has been demonstrated previously (*Britton, Fat & Neligan, 2020*; *Hu et al., 2020*). Given this limitation, there is a need to evaluate whether alcohol volume is associated with sleep health during the pandemic. Such research would have important implications regarding public health messaging with respect to sleep health.

Several limitations must be considered. First, a cross-sectional design was used to examine sleep health and associated lifestyle behaviours in Australian community-level tennis players. This approach does not enable examination of sleep health and lifestyle behaviours prior to the pandemic. Therefore, it is not possible to ascertain whether COVID-19 negatively impacted sleep health and lifestyle behaviours. Second, alcohol consumption during the pandemic was examined using specific response items as opposed to millilitres or number of standard drinks per week and occasion (to detect binge drinking), which likely led to a ceiling effect with respect to participant responses and the inability to draw strong conclusions between alcohol consumption and sleep health. Third, this survey was disseminated *via* social media, which might have led to a selection bias and, therefore, our sample may not be representative of community-level tennis players. Lastly, while this study included respondents from all states of Australia, their specific location was not collected. This is of relevance as government-imposed restrictions slightly differed between states, which may have led to a state-specific effect with respect to sleep health and lifestyle behaviours during the pandemic.

This study's strengths must also be acknowledged. The distributed survey included validated sleep health questionnaires (Sleep Health Index, Sleep Satisfaction Tool), which were developed by the National Sleep Foundation. The use of these specific questionnaires allowed for the assessment of multiple dimensions of sleep health, as well as the comparison to normative values. Further, given the homogenous sample of this study, 285 respondents, should be considered high.

## CONCLUSIONS

This is the first study to investigate sleep health and its associations with lifestyle behaviours during the COVID-19 pandemic in adult Australian community-level tennis players. These findings herein are of interest as they provide a unique perspective into the sleep health and lifestyle behaviours of a physically active population during the initial lockdown period within Australia. Tennis players had better sleep health outcomes compared to existing normative values, however sleep health outcomes were not related to tennis play or physical activity level, indicating that other unexplored factors, such as financial status and socialisation may have influenced sleep health. Future studies should explore the longitudinal link between sleep health and the aforementioned factors in sporting populations, including community tennis players.

## ACKNOWLEDGEMENTS

The authors would like to thank all tennis clubs, coaches, and players in Australia for disseminating the online survey. Additionally, the authors would like to thank everyone that completed the survey.

### Funding

The authors received no funding for this work.

### Competing Interests

The authors declare that they have no competing interests.

### Author Contributions

- Philipp Beranek conceived and designed the experiments, performed the experiments, analyzed the data, prepared figures and/or tables, authored or reviewed drafts of the paper, and approved the final draft.
- Travis Cruickshank conceived and designed the experiments, performed the experiments, analyzed the data, prepared figures and/or tables, authored or reviewed drafts of the paper, and approved the final draft.
- Olivier Girard analyzed the data, prepared figures and/or tables, authored or reviewed drafts of the paper, and approved the final draft.
- Kazunori Nosaka analyzed the data, prepared figures and/or tables, authored or reviewed drafts of the paper, and approved the final draft.

- Danielle Bartlett analyzed the data, prepared figures and/or tables, authored or reviewed drafts of the paper, and approved the final draft.
- Mitchell Turner conceived and designed the experiments, performed the experiments, analyzed the data, prepared figures and/or tables, authored or reviewed drafts of the paper, and approved the final draft.

## Human Ethics

The following information was supplied relating to ethical approvals (*i.e.*, approving body and any reference numbers):

This study was approved by the Edith Cowan University Human Research Ethics Committee (2020-01367).

## Data Availability

The raw data is available in the Supplemental File.

## Supplemental Information

Supplemental information for this article can be found online at http://dx.doi.org/10.7717/peerj.13045#supplemental-information.

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
