# Peer review of "Sleep health of Australian community tennis players during the COVID-19 lockdown"

_PeerJ, doi:10.7717/peerj.13045_

## Round 0.1 · original submission · Major Revisions

Kindly justify clearly the link with COVID-19 and pre-COVID situation.

·

Basic reporting

The manuscript is clear, unambiguous, used professional english throughout. In detail background is provided using references from literature. Article is structured, figures, tables and raw data is also shared.

Experimental design

It is a original piece of research within aims and scope of the journal. This research also stated how it fills an identified knowledge gap. rigorous investigation performed to a high technical and ethical standard. Methodology section was so clear with sufficient detail and information to replicate.

Validity of the findings

All underlying data have been provided, they are robust, statistically sound.

Conclusion needs a minor revision, it is just like a summary of the research. Conclusions should be supported by the results obtained not just the summary of findings.
Are the implications of the study identified?
Are there any recommendations for further research?
Does those recommendations identify how weaknesses in the study design could be avoided in future research?

Additional comments

comment 1 - In abstract, methods section would be better if authors can add study design and sample size.
comment 2 - only limitations of the study mentioned, that would be great if the strengths of the study is also provided.

·

Basic reporting

It is suggested that in lines 42 - 43 you only focus on the results of the study. Avoid making conclusions that are beyond the scope of the research.

The introduction section of the article needs some more literature reviews on sleep normative data.

Improve the sentence in lines 83 - 84: “Furthermore, while the pandemic was still ongoing and most facilities continued to be closed, tennis was one of the first sports able to recommence given the”

Experimental design

How did you ensure that participants only played tennis during the Pandemic? Give some more information on this in line 111.

In the method section, include the psychometric properties of the questionnaires provided.

In the Data Analysis section: Provide analysis on the association between age and sleep health.

Validity of the findings

Clearly define the restrictions imposed by the government in lines 256 - 258. Was there a total lockdown?

Your research is not focused on employment. Consider explaining only the results of the research in lines 270 - 272.

Additional comments

No comments.

·

Basic reporting

The paper is well written and describes the findings of a cross-sectional survey in Australian community tennis players during the COVID-19 lockdown 2020. Main variables are sleep health, additional variables are tennis play, training location, physical activity and alcohol consumption. The authors found sleep health to be generally good among tennis players if compared to normative values from the literature. They did not find any correlation of sleep health with tennis play, physical activity, training location and alcohol consumption. These findings are important since they contradict other studies that found worsening of sleep during the pandemic in the general public and in athletes.
Important literature has been cited with few exceptions mentioned below. I have some issues with the figure and the tables mentioned below.

Abstract:
- Line 38: The Tb- should be Tb.

Introduction:
- Since you measured alcohol consumption please insert a short paragraph on the current data on alcohol consumption during COVID-19 lockdowns in (1) the general public and (2) athletes
- Please insert a short paragraph on the impact of lockdown on sports facilities and training hours of athletes.

Table 1:
I find the current depiction of results difficult to read. I would suggest adding the % behind respective values and leaving out the “n =”. You might want to rewrite the table with the corresponding description (% and n ; median and interquartile range) in the head-row of the table and inserting the subtitles (Age range, relationship status and so on) below. This would increase readability.

Table 2:
To improve readability I would suggest brackets around the p-values. This way it is easier to distinguish coefficients (Tb) from p-values.

Figure 1:
Please indicate what is depicted by the thin black curved line around the boxplots.

Experimental design

The methods are well described, with sufficient detail to replicate. No additional comments

Validity of the findings

These findings are important since they contradict other studies that found worsening of sleep during the pandemic in the general public and in athletes. I have some comments on the discussion section:

Discussion:
- Lines 240-241, it is complicated to see which values are the ones from the tennis players and which the normative values. Please improve this, I would suggest: “SHI (normative: 76 vs. tennis players: 85.3)”
- Lines 267-269: I don’t think your findings contradict an association between physical activity and sleep since the vast majority in your sample remained physically active and met WHO recommendations. Poor sleep is strongly associated with sedentary behavior. Therefore, your sample already consists of participants with higher physical activity level than the general public.
- Lines 278-281: One hour of outdoor training per week has no effect on zeitgebers. Therefore, this line of argument is frail. I would suggest excluding it or adapting it in order to be more appropriate.
- I agree that the responses on alcohol consumption are not suitable to detect more heavy drinking patterns and this is an important limitation of your findings. However, standard drinks per week and per occasion (to detect binge drinking) would have been more appropriate (not millilitres).
- Since questionnaires were distributed by social media you have no representative sample of community tennis players. Selection bias might have occurred. Please state this as a limitation.

Additional comments

I don’t see a paragraph on funding. I would suggest adding this to the manuscript.

---

## Round 0.2 · accepted · Accept

Thanks for making the changes.